# Coordination among neighbors improves the efficacy of Zika control despite economic costs

Natalie J. Lemanski[1¤]*, Samantha R. Schwab[2], Dina M. Fonseca[3], Nina H. Fefferman[1]

**1** Department of Ecology and Evolutionary Biology, University of Tennessee, Knoxville, Tennessee, United States of America, **2** Department of Ecology, Evolution, and Natural Resources, Rutgers University, New Brunswick, New Jersey, United States of America, **3** Center for Vector Biology, Department of Entomology, Rutgers University, New Brunswick, New Jersey, United States of America

¤ Current address: Department of Ecology and Evolutionary Biology, University of California, Los Angeles, California, United States of America
* Natalie.lemanski@gmail.com

**Data Availability Statement:** All model code is available in the Dryad depository, https://doi.org/10.5068/D1366B.

## Abstract

Emerging mosquito-borne viruses like Zika, dengue, and chikungunya pose a major threat to public health, especially in low-income regions of Central and South America, southeast Asia, and the Caribbean. Outbreaks of these diseases are likely to have long-term social and economic consequences due to Zika-induced congenital microcephaly and other complications. Larval control of the container-inhabiting mosquitoes that transmit these infections is an important tool for mitigating outbreaks. However, metapopulation theory suggests that spatiotemporally uneven larvicide treatment can impede control effectiveness, as recolonization compensates for mortality within patches. Coordinating the timing of treatment among patches could therefore substantially improve epidemic control, but we must also consider economic constraints, since coordination may have costs that divert resources from treatment. To inform practical disease management strategies, we ask how coordination among neighbors in the timing of mosquito control efforts influences the size of a mosquito-borne infectious disease outbreak under the realistic assumption that coordination has costs. Using an SIR (Susceptible-Infectious-Recovered)/metapopulation model of mosquito and disease dynamics, we examine whether sharing surveillance information and coordinating larvicide treatment among neighboring patches reduces human infections when incorporating coordination costs. We examine how different types of coordination costs and different surveillance methods jointly influence the effectiveness of larval control. We find that the effect of coordination depends on both costs and the type of surveillance used to inform treatment. With epidemiological surveillance, coordination improves disease outcomes, even when costly. With demographic surveillance, coordination either improves or hampers disease control, depending on the type of costs and surveillance sensitivity. Our results suggest coordination among neighbors can improve management of mosquito-borne epidemics under many, but not all, assumptions about costs. Therefore, estimating coordination costs is an important step for most effectively applying metapopulation theory to strategies for managing outbreaks of mosquito-borne viral infections.

**Funding:** This work was supported by the National Science Foundation (www.nsf.gov) under grant no. DEB-1640951 (awarded to NF). The funders had no role in study design, data collection and analysis, decision to publish, or preparation of the manuscript.

**Competing interests:** The authors have declared that no competing interests exist.

## Author summary

Mosquito-borne viruses, such as Zika, are an urgent public health threat, particularly in tropical, low-income regions. Vector control, the main strategy for combatting outbreaks, can be challenging because the urban-adapted, container-breeding mosquitoes that transmit these viruses often exhibit metapopulation dynamics, where mortality in one population is compensated by migration from neighboring populations. The timing and spatial distribution of vector control efforts can therefore have a large impact on their efficacy. Using a model of virus transmission and vector population dynamics, we demonstrate that local mosquito control initiatives aimed at reducing the burden of Zika and other mosquito-borne infections are most effective when there is communication of surveillance findings among neighboring control agencies and coordination over the timing of mosquito reduction treatments. We find that local communication improves epidemic outcomes even when it imposes costs to resource-limited control agencies due to gains in the efficiency of mosquito control from spatial coordination.

## Introduction

Recent outbreaks of mosquito-borne viruses such as Zika [1], chikungunya [2], and dengue [3] are a major public health threat in tropical regions, placing a large burden on the healthcare systems of affected countries. Dengue virus is responsible for nearly 10,000 deaths and 60 million infections annually [4], while the ongoing Zika epidemic will likely have long-term public health consequences due to Zika-induced congenital microcephaly in fetuses and Guillain-Barré syndrome in adults [5–8]. As climate change and increased human migration facilitate the spread of these diseases to new locations [9], there is an urgent need to respond quickly to outbreaks while making efficient use of the limited resources available to combat them. It is therefore vital to develop methods for predicting the extent and severity of these epidemics [9–12] and how different public health interventions will impact them [13].

Emerging or reemerging infections often have no vaccine available yet (as with Zika virus [14,15]) or have vaccines with only limited availability and/or efficacy (as with dengue virus [5,16]). Vector control is therefore the most widely used strategy for limiting the size and severity of outbreaks [17]. Two main vectors for these viruses are *Aedes aegypti* and *Ae. albopictus* [18], both widespread urban-adapted mosquitoes whose immature larvae develop in small, ephemeral water sources, such as planters and trash cans [19]. Several strategies for controlling these mosquito populations have been used in arbovirus-endemic regions [20,21]. Mosquito control programs often use chemical larvicide treatment or container removal (i.e. source reduction) to reduce larval survival because it is the easiest life stage to target [13,22]. Source reduction is labor-intensive, however, with species like *Aedes* that oviposit in small, ephemeral pools because these sites are numerous and often cryptic [23]. Adulticide is widely used as well, either alone or in combination with larvicide [13,20,21,24]. The organization of vector control programs varies from top-down, centralized approaches, such as aerial spraying of insecticides, to community-based initiatives that provide education and/or resources to help local residents reduce mosquito survival at the household level [13,20,21,25]. Controlled studies of current mosquito control programs have found wide variation in their efficacy [20]. Given the cost and resource intensiveness of these programs, it is important to be able to predict which types of interventions are most likely to be effective at reducing arbovirus infections.

Epidemiological models of these viruses must consider how vector habitat is often short-lived and patchily distributed across the landscape [17]. This type of habitat is prone to meta-population dynamics, in which individual sub-populations frequently go extinct and are recolonized by adult migration [26]. Although *Aedes* adults disperse over short distances [27–29], control measures that are conducted at the neighborhood level [e.g. 23] may lead to rescue effects in which untreated patches act as source populations for adjacent treated patches [26]. If juvenile population growth displays density dependence within patches [30], migration of biting adults from untreated patches may compensate for mortality in treated patches, diminishing the effectiveness of disease control. The synchronicity of treatment across patches may therefore also be important, with patches treated at different times allowing untreated patches to act as reservoirs, increasing the potential for rescue effects relative to cases in which all patches are treated at once.

Metapopulation theory suggests that more uniform vector control across a landscape should reduce the size of an outbreak better than patchy control [31]. Previous work also indicates the spatial distribution of patch-level control can influence its effectiveness at mitigating an outbreak [32]. In practice, though, control is often implemented at the local level by individual communities or households [33,34], without consideration of the best timing or spatial distribution for landscape-level control [35]. Given these limitations, coordination of vector control among patches has the potential to greatly improve the efficiency of disease mitigation during an outbreak [36].

However, coordination between local mosquito control agents may also incur its own costs. In the worst cases, such costs may decrease control efforts by requiring the reallocation of resources to enable the coordination itself. The largest disease burdens from *Aedes*-borne viruses, including Zika, dengue, and chikungunya, occur in resource-limited regions, such as southeast Asia and Latin America [4,5]. Money spent on communication and coordination among households or neighborhoods is likely to reduce the money available for implementing control.

Costs can also take the form of time delays since time spent communicating with other regions or waiting for other regions to be ready for coordinated treatment can delay the start of treatment in an affected patch. Previous findings suggest that surveillance resulting in earlier treatment after an outbreak is most effective in reducing human infections [37]. Even if coordinated treatment has the potential to improve disease control, the reduction in efficacy from delaying treatment may outweigh the benefits from coordination.

The questions we therefore seek to answer in the present work are: a) under what circumstances do the benefits of coordinating larval mosquito control among local patches outweigh the costs and b) what level of coordination best reduces the number of human infections during an outbreak?

To answer these questions, we build on a spatially explicit SIR (Susceptible-Infectious-Recovered) model first presented in [37]. We examine two different possible costs of coordination: costs in time or costs in resources. We then examine three different coordination scenarios: no coordination, coordination with nearest geographic neighbors, and coordination with both nearest and second nearest neighbors to determine the optimal level of coordination, given different assumptions about the method of surveillance used to determine treatment decisions and the costs of coordination among patches.

## Methods

We assume the landscape is divided into a simple 5 by 4 grid of identical habitat patches, where patch size is defined as the scale over which adult mosquitoes migrate. Because of the

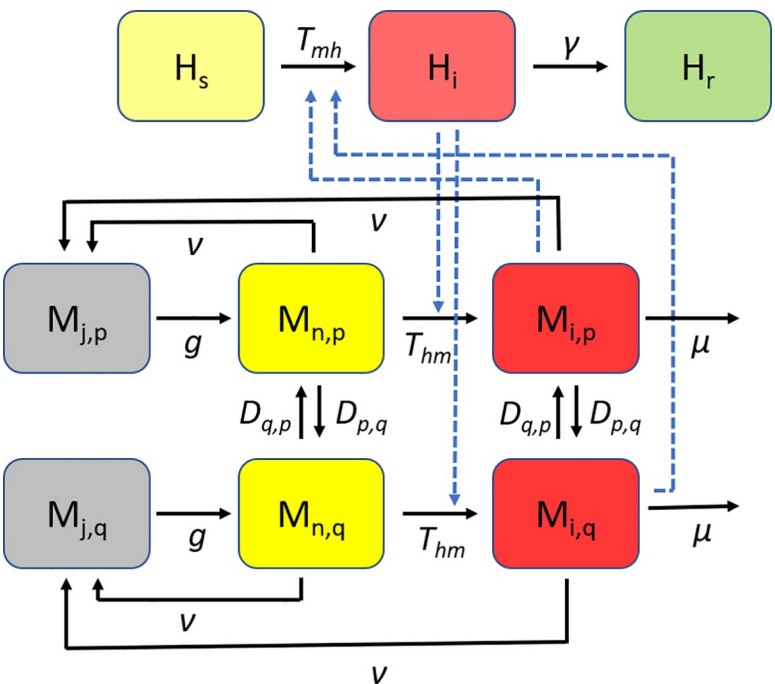

**Fig 1. Conceptual diagram of compartmental (SIR) model.** Solid lines indicate transitions between model compartments. Dashed lines indicate dependencies. Susceptible humans ($H_s$) become infected ($H_i$) at rate $T_{mh}$ when bitten by infected adult mosquitoes ($M_i$). Infected humans recover at rate $\gamma$ and become resistant to infection ($H_r$). Uninfected adult mosquitoes ($M_n$) become infected ($M_i$) at rate $T_{hm}$ by biting an infected human. Juvenile mosquitoes in each patch ($M_{j,p}$) are produced from adult mosquitoes in that patch ($M_{n,p}$ and $M_{i,p}$) at a density-dependent rate, $v$, and mature to become uninfected adult mosquitoes ($M_{n,p}$) at rate $g$. Adult mosquitoes in each patch, $p$, migrate to each adjacent patch, $q$, at rate $D_{p,q}$ and migrate from each adjacent patch, $q$, at rate $D_{q,p}$. Adult mosquitoes die and are removed from the population at rate $\mu$.

relatively short flight range of adult *Aedes* mosquitoes [38], we assume that mosquitoes disperse only among adjacent patches, with equal probability (S1 Table). We assume humans are mobile enough at the spatial scale of the model that a mosquito in any patch can bite any human (but see [36] for analyses in which this assumption is relaxed). The time scale of the model is such that we assume a fixed human population (no human birth or death).

Mosquito population dynamics and disease transmission dynamics are described by a series of discrete time, SIR-type difference equations (Fig 1). These equations are the same as in [37] (reproduced in S1 Text for ease of readership). Model variables are listed in S2 Table. During the period of surveillance and control, participating patches are treated the following day when a threshold for control is exceeded (S1 Table). For simplicity, we assume that all mosquito control targets larvae (larvicide or source reduction), a common form of control for container-inhabiting mosquitoes [13,39,40]. When a patch is treated, its larval mosquito population is reduced for ten days (adult mosquitoes are not directly affected; S1 Table). Although in practice, mosquito control efforts may include a combination of larvicide and adulticide treatments [13,21], in order to isolate the effects of larvicide treatment in our model, we assume that no adulticide is used. For full list of model assumptions, see Table 1.

We examined two different types of surveillance data as the trigger for implementation of larval control: total adult mosquitoes per patch and infected adult mosquitoes per patch. We used the surveillance metric of total adult mosquitoes per patch because previous work found that was the most effective surveillance tactic [37]. We also used the metric of infected adult mosquitoes per patch to compare the effects of coordination when surveillance is based on

**Table 1. Assumptions of the model.**

| Topic | Assumptions |
|---|---|
| Landscape | All patches are identical and have the same carrying capacity. |
| | Mosquitoes can disperse only to adjacent patches and are equally likely to disperse in any direction. |
| | Humans move through the landscape and are equally likely to be bitten in any patch. |
| Control | Surveillance results are 100% accurate and can be obtained fast enough to inform control actions the following day. |
| | Larvicide treatment is effective for 10 days and results in 100% larval mortality (though see S1 and S2 Figs for effects of lower treatment efficacy). |
| | Larvicide is the only control measure implemented (no adulticide). |
| Epidemiology | The virus is only transmitted horizontally, between humans and mosquitoes. |
| | Recovery in humans results in lifelong immunity from infection. |
| | Infected mosquitoes do not recover from infection. |
| | Viral infection has no effect on mosquito mortality, fecundity, or dispersal. |
| Population | No evolution occurs in the mosquito population over the model timespan. |
| | No human birth or death occurs over the model timespan. |
| | Adult mosquito dispersal and survival are not density-dependent. |
| | Larval mosquito survival is density-dependent. |

demographic vs. epidemiological information (see [37] for description of other surveillance methods.)

For each surveillance method, we also examined two different treatment thresholds, since previous work indicated that the sensitivity of the treatment threshold can influence the relative effectiveness of different methods of mosquito surveillance [37]. In the high sensitivity condition, the thresholds for treatment were: one infected mosquito for the epidemiological-based surveillance, or 10% of mean pre-control adult mosquito abundance for demographic-based surveillance. In the low sensitivity condition, the thresholds were: five infected mosquitoes for the epidemiological-based surveillance, or 50% of mean pre-control adult mosquito abundance for demographic-based surveillance (S1 Table). Model implementation and analyses were performed in Matlab [41].

## Types of coordination

We define a patch's nearest neighbors as all patches orthogonally and diagonally adjacent to it (adult mosquito dispersal occurs only between patches that are nearest neighbors). We define a patch's second nearest neighbors as all patches adjacent to its nearest neighbors. For each combination of surveillance method and treatment threshold, we examined three types of coordination. In the no coordination condition, each patch was treated with larvicide the day following when its own threshold for control was exceeded (either the number of adult mosquitoes or the number of infected mosquitoes, depending on the type of surveillance).

In the nearest neighbor condition, when a patch's threshold for control was exceeded, the focal patch was treated the next day and its nearest neighbors were treated one day later. In the second nearest neighbor condition, when a patch's threshold for control was exceeded, the focal patch, its nearest neighbors, and its second nearest neighbors were all treated, with nearest neighbors treated the day after the focal patch and second nearest neighbors treated one day after that.

## Types of costs

The effect of coordination among patches on the efficiency of control may depend on the types of costs associated with coordination. We examined three alternate assumptions about

the costs of coordination: no costs, costs in time, and costs in resources. In the "no costs" scenario, we assume coordination causes no change in the timing or efficacy of larval control. In the "costs in time" scenario, we assume coordination causes delays in treatment. In the "costs in resources" scenario, we assume that coordination reduces larvicide efficacy in each treated patch.

Coordinating with nearby patches may delay control implementation because it takes time for jurisdictions to communicate their surveillance findings with each other. In addition, staff employed in coordination efforts may reduce staff available for control, resulting in additional delays (even if it does not otherwise reduce efficacy). Under this "costs in time" scenario, we assumed that the greater the scale of coordination, the greater the delay it causes. Thus, nearest neighbor coordination causes control in all treated patches to be delayed by one additional day and second nearest neighbor coordination causes control in all treated patches to be delayed by two additional days.

If the agencies or individuals responsible for control have limited financial resources, money spent on coordination may mean less money available to spend on actual treatment. There may therefore be a trade-off between treatment efficacy (defined as the fraction of larvae killed by treatment) and the scale of coordination. In previous work [37], it was assumed that treatment is 100% effective (all larvae in a treated patch are killed). Under this "costs in resources" scenario, we assume that there is 100% treatment efficacy only in the absence of coordination, and that coordination reduces efficacy, with greater scales of coordination causing greater reductions (S1 Table). A sensitivity analysis indicated that an initial treatment efficacy less than 100% does not substantially alter our results (S1 and S2 Figs). Although the magnitude of the costs in our simulation were arbitrarily chosen, our results should give insight into whether costs of coordination influence the optimal control strategy and therefore whether empirical estimation of costs is useful to inform control.

For each combination of surveillance type, treatment threshold, and cost scenario, we performed 100 simulation runs. In keeping with previous work, for each simulation run, 16 out of 20 patches were chosen randomly to participate in surveillance and treatment [37]. This is to simulate the realistic scenario in which not all patches can participate in surveillance for reasons unrelated to mosquito or disease dynamics (such as inaccessibility or private land). We assume that the goal of mosquito control is to reduce the number of human infections during an epidemic; thus, for each run, we calculated the percent reduction in human infections compared to the case in which no control is implemented.

## Results

### Surveillance based on number of adult mosquitoes

We find that when surveillance is demographic (based on the number of adult mosquitoes), the optimal level of coordination for outbreak mitigation depends on both the sensitivity of the vector control threshold and the type of coordination costs. When demographic surveillance is highly sensitive and there are no costs associated with coordination, vector control in response to the nearest neighbors' treatment thresholds being triggered has a very small benefit compared to no coordination (Fig 2). Coordination with second nearest neighbors has no additional benefit over coordination with only the nearest neighbors. In contrast, when surveillance is less sensitive and there are no costs of coordination, coordinating with nearest neighbors reduces human infections more than not coordinating. Coordinating with second nearest neighbors reduces human infections even further (Fig 2).

When demographic surveillance is highly sensitive, and coordination causes a delay in treatment, we find that coordinating with nearest neighbors has little effect on epidemic

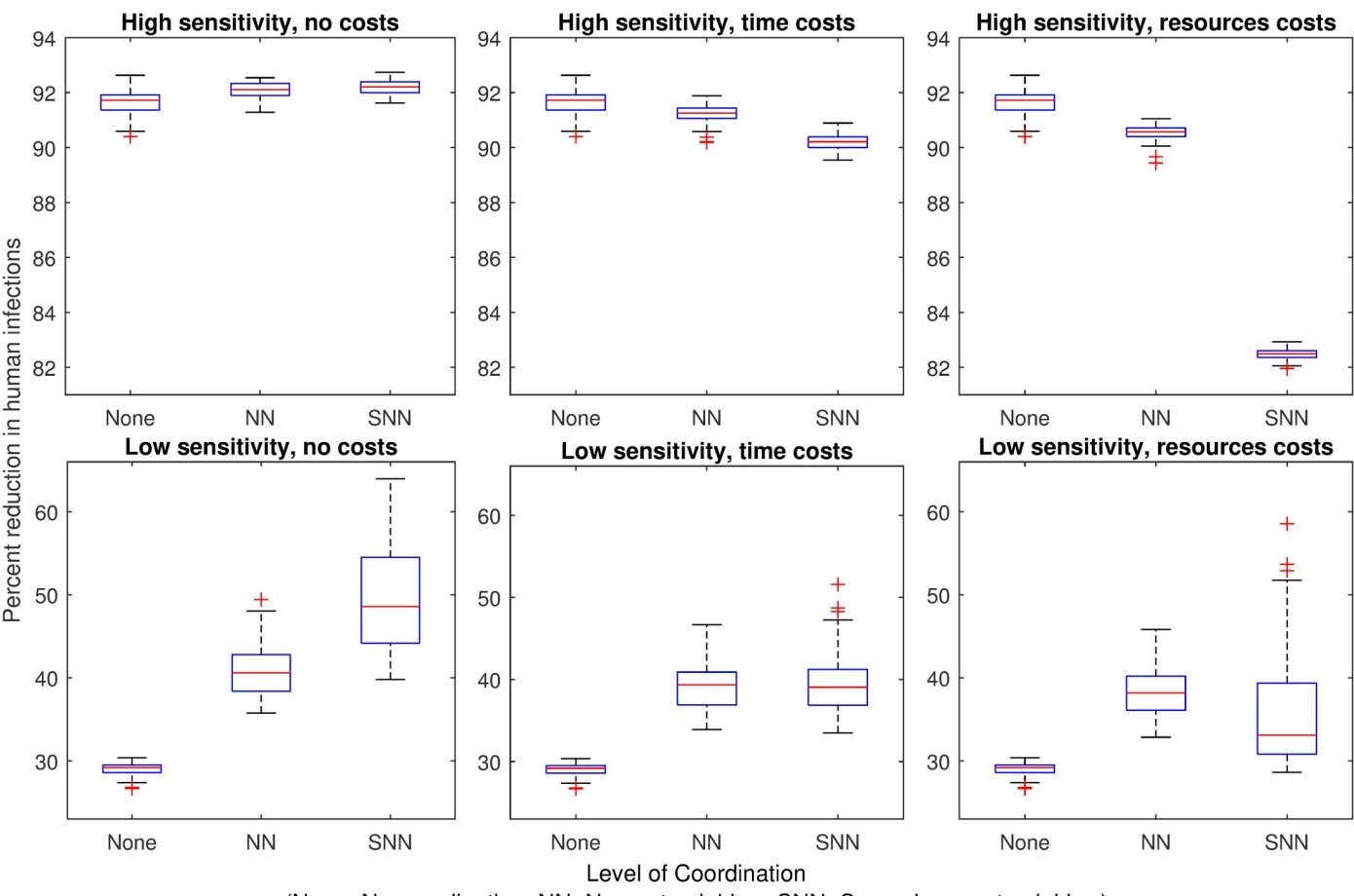

**Fig 2. Effect of mosquito control on human infections when surveillance is based on mosquito abundance.** Panels 1–6 show the percent reduction in total human infections during an outbreak, for each mosquito control scenario, relative to the number of infections that occur when no mosquito control is implemented. Mosquito control occurs, reducing juvenile mosquitoes in a patch, when the number of adult mosquitoes in the patch exceeds a control threshold. With no coordination ("None"), control occurs in a single patch only when its own threshold is exceeded. With nearest neighbor coordination ("NN"), control occurs in a patch when any adjacent patch's threshold is exceeded. With second nearest neighbor coordination ("SNN"), control occurs in a patch when the threshold is exceeded in any patch adjacent to the focal patch's nearest neighbors. In the high sensitivity scenarios (top panels), the control threshold = 10% average pre-control abundance; in the low sensitivity scenarios (bottom panels), the control threshold = 50% average pre-control abundance. In the "no costs" scenario (left panels), control in response to neighbors' surveillance information has no additional costs. In the "time costs" scenario (central panels), coordination with first or second nearest neighbors delays treatment for one or two days, respectively. In the "resource costs" scenario (right panels), coordination with neighbors reduces the efficacy of treatment (juveniles are reduced by 90% or 80% instead of 100%).

control outcomes compared to not coordinating (Fig 2). Furthermore, coordinating with second nearest neighbors has worse outcomes than not coordinating at all. In contrast, when surveillance is less sensitive, coordination with nearest neighbors improves epidemic outcomes even when coordination causes a delay. Coordinating with second nearest neighbors also improves outcomes over not coordinating but has no additional benefit over coordinating only with nearest neighbors (Fig 2).

When we assume that coordination is costly in resources (leading to a reduction in larvicide efficacy), we find that for highly sensitive demographic surveillance, coordination with nearest neighbors leads to worse control outcomes than no coordination (Fig 2). Furthermore, coordinating with second nearest neighbors leads to much worse outcomes than either no coordination or coordination with only nearest neighbors. However, when coordination is costly in resources and surveillance is less sensitive, coordinating with nearest neighbors is better than

not coordinating. Coordinating with second nearest neighbors also improves outcomes over not coordinating but not as much as only coordinating with nearest neighbors (Fig 2). Though we assumed in this model that larvicide treatment is 100% effective in the absence of coordination (S1 Fig), a sensitivity analysis found that treatment efficacy can affect the benefits of coordination (S1 Fig). When surveillance has low sensitivity and treatment results in less than ~70% larval mortality, any further reduction in efficacy resulting from coordination costs eliminates the benefits of coordination.

## Surveillance based on number of infected mosquitoes

When surveillance is epidemiological (based on the number of infected mosquitoes) rather than demographic, we find that the effect of coordination depends on the type of costs but not the sensitivity of detection. Without any costs of coordination, coordinating with nearest neighbors achieves a greater reduction in human infections than not coordinating (Fig 3). Coordination with second nearest neighbors has little or no additional benefit over coordination with nearest neighbors.

When coordination is costly in time, coordination with nearest neighbors achieves the greatest reduction in human infections (Fig 3). Coordinating with second nearest neighbors also achieves greater infection reduction than not coordinating at all but has less benefit than only coordinating with nearest neighbors (Fig 3). When coordination is costly in resources, coordinating with nearest neighbors still achieves the best control outcomes, but coordinating with second nearest neighbors leads to worse infection outcomes than not coordinating at all (Fig 3). In addition, when the initial larvicide efficacy is less than 100%, any reduction in efficacy as a result of coordination costs results in worse outcomes than not coordinating (S2 Fig).

These results suggest that when surveillance is epidemiological and larvicide is highly effective, the optimal level of coordination is nearest neighbors, regardless of costs or sensitivity. Inclusion of second nearest neighbors does not substantially improve infection reduction and can even detract from infection reduction, depending on the coordination costs. It is also important to note that epidemiological surveillance was much less effective than demographic surveillance, regardless of the type of coordination employed (Figs 2 and 3).

## Discussion

Emerging mosquito-borne infectious diseases like Zika, chikungunya, and dengue represent an urgent public health threat that is likely to grow in severity as climate change facilitates host range expansion [9]. Population control of the mosquito vectors that transmit these infections remains the main tool for mitigating epidemics. However, the often cryptic and patchily distributed habitat in which these mosquitoes oviposit makes control resource-intensive [23] and makes it difficult for all patches in the landscape to participate in surveillance and treatment.

In addition, the greatest disease burden of many arboviruses occurs in resource-limited regions of Latin America and southeast Asia [4,42] where there may be limited staff and funds available for vector control. It is therefore important to allocate surveillance and control efforts efficiently to maximize their effect on reducing human infections.

Because of metapopulation dynamics, coordinating the timing of habitat treatment across the landscape has the potential to improve the efficiency of control [36,37]. However, if coordination efforts require time and money, it is important to know under what circumstances coordination is worthwhile and when it is instead better to allocate those resources elsewhere.

Our simulations suggest that the ideal level of coordination in vector control efforts among neighboring patches depends on the type of cost associated with coordination, as well as both the type and sensitivity of surveillance on which treatment decisions are based. When control agencies can enact highly sensitive surveillance based on the number of adult mosquitoes,

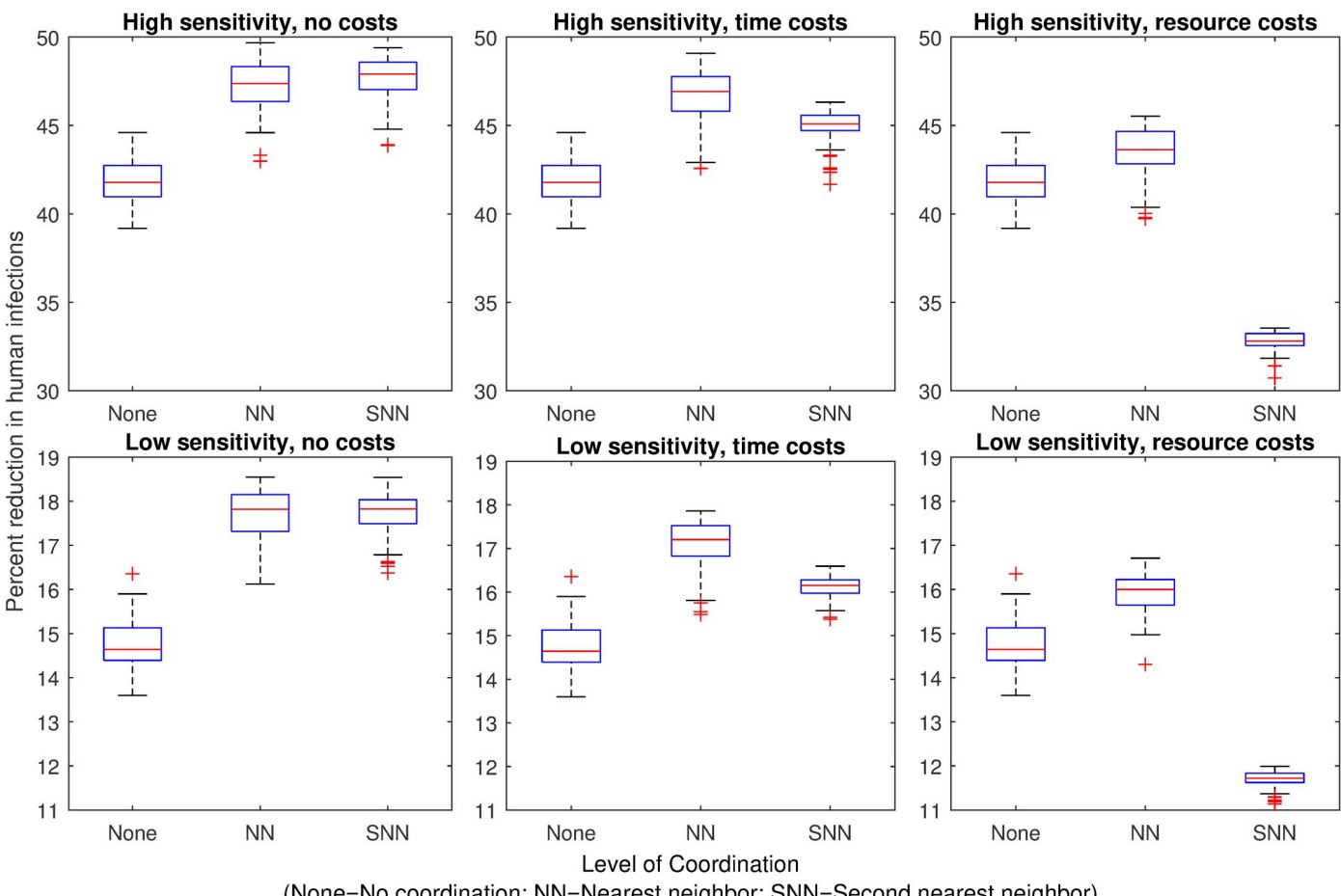

**Fig 3. Effect of mosquito control on human infections when surveillance is based on mosquito infections.** Panels 1–6 show the percent reduction in total human infections during an outbreak, for each mosquito control scenario, relative to the number of infections that occur when no mosquito control is implemented. Mosquito control occurs when the number of infected mosquitoes in a patch exceeds the control threshold. With no coordination ("None"), control occurs in a single patch only when its own threshold is exceeded. With nearest neighbor coordination ("NN"), control occurs in a patch when any adjacent patch's threshold is exceeded. With second nearest neighbor coordination ("SNN"), control occurs in a patch when the threshold is exceeded in any patch adjacent to the focal patch's nearest neighbors. In the high sensitivity scenarios (top panels), the control threshold = 1 infected mosquito; in the low sensitivity scenarios (bottom panels), the control threshold = 5 infected mosquitoes. In the "no costs" scenario (left panels), coordination with neighbors has no additional costs. In the "time costs" scenario (central panels), coordination with first or second neighbors delays treatment for one or two days, respectively. In the "resource costs" scenario (right panels), coordination with neighbors reduces the efficacy of treatment (juveniles are reduced by 90% or 80% instead of 100%).

there is little to be gained from coordination, and even small costs in either time or resources are likely to outweigh the benefits. This is mostly because at high sensitivity of detection, very high reduction in infections can be accomplished without coordination among patches. However, this level of sensitivity most likely represents an uncommon, best-case scenario, particularly in low income countries.

When demographic surveillance is less sensitive or when control decisions are based on surveillance of infected mosquitoes, coordination between adjacent patches can substantially improve epidemic control outcomes compared to each patch enacting control independently. This benefit of coordination persists even when it causes a delay in treatment or a reduction in resources available for larval control. When surveillance is epidemiological, the ideal spatial scale for coordination of mosquito control is the scale over which adult mosquitoes migrate (adjacent patches). When surveillance is demographic, coordinating on a larger spatial scale can be beneficial if it does not reduce the efficacy of treatment within patches. While we

assumed in this study that larvicide treatment in completely effective, assuming a lower effectiveness did not substantially alter our conclusions, except when surveillance was based on total mosquito abundance and less sensitive, and when coordination was costly in resources. In that case, a lower initial efficacy meant that any reduction in efficacy resulting from coordination costs resulted in more human infections than no coordination among patches.

Previous work has found that larval control based on surveillance of adult mosquito numbers is much more effective at reducing human infections than control based on surveillance of mosquito infections [37]. Our results show that even with coordination, epidemiological surveillance does not achieve the same reduction in human infections as demographic surveillance (Figs 2 and 3). However, in realistic epidemic scenarios, funding for control may not be available until after an infection is introduced, particularly in resource-limited areas [21]. In addition, chemical larvicide may have environmental side effects that must be balanced against disease prevention. While we assumed in this study that surveillance focused on adult mosquitoes, surveillance of larval mosquito numbers is used as well [21] and previous work suggests that the sensitivity of surveillance information can influence which surveillance target is best for reducing human infections [37]. Given that vector control in response to highly sensitive demographic surveillance is not always possible, coordination of vector control has the potential to improve epidemic mitigation under a broad range of realistic surveillance scenarios.

For simplicity, the present work focuses on larval control, since that is the life stage commonly targeted in local and community-based mosquito control efforts [13,39,40]. However, it is important to note that mosquito control efforts may involve a combination of larvicide and adulticide [13,24]. Research examining how targeting adults vs. larvae will influence the efficacy of mosquito control at mitigating an arboviral epidemic is currently underway.

## Conclusions

Our findings suggest that coordination in surveillance and control among neighboring patches can improve arboviral epidemic outcomes under a range of realistic circumstances, even when costs of coordination are considered. However, the nature of costs and the sensitivity of surveillance are both important factors in the benefit of coordination and the ideal scale over which to communicate surveillance findings.

While we did not examine the effect of varying the magnitude of coordination costs, our findings suggest that it is likely to be important in assessing under what conditions coordination will be beneficial. It would be useful for future theoretical work to examine the threshold costs beyond which coordination ceases to be worthwhile under various circumstances. Overall, our findings suggest that greater coordination among local vector control agents has significant potential to reduce the severity of mosquito-borne epidemics, but accurate estimations of the costs associated with coordination are needed to inform best practices for mosquito control implementation.

## Supporting information

**S1 Text. This methods appendix contains the full set of SIR-type difference equations we used to model disease dynamics and mosquito population dynamics.** These equations are adapted from [37].
(DOCX)

**S1 Table. This table contains the name and definition of each parameter used in the model, along with the values used in our model simulations.**
(DOCX)

**S2 Table. This table contains the name and definition of each variable that appears in the model.**
(DOCX)

**S1 Fig. Effect of treatment efficacy on the benefits of coordination when control is based on surveillance of adult mosquitoes.** Blue lines (None) show the reduction in human infections when larval control is enacted only in patches that exceed the surveillance threshold. Red lines (NN) show the result when control occurs in the triggered patch and its nearest neighbors. Yellow lines (SN) show the result when control occurs in a triggered patch's nearest neighbors and second nearest neighbors. At high surveillance sensitivity, the efficacy of larvicide treatment has no impact on the benefits of coordination among neighbors. At low sensitivity, the efficacy of the larvicide treatment does affect the benefits of coordination but only when coordination reduces the treatment efficacy in each patch (right panels).
(TIF)

**S2 Fig. Effect of treatment efficacy on the benefits of coordination when control is based on surveillance of infected mosquitoes.** Blue lines (None) show the reduction in human infections when larval control is enacted only in patches that exceed the surveillance threshold. Red lines (NN) show the result when control occurs in the triggered patch and its nearest neighbors. Yellow lines (SN) show the result when control occurs in a triggered patch's nearest neighbors and second nearest neighbors. Without costs, coordination with nearest neighbors improves infection outcomes. However, unless larvicide has very high initial efficacy, any reduction in efficacy resulting from coordination results in worse outcomes than not coordinating.
(TIF)

## Author Contributions

**Conceptualization:** Natalie J. Lemanski, Samantha R. Schwab, Nina H. Fefferman.

**Formal analysis:** Natalie J. Lemanski, Samantha R. Schwab.

**Funding acquisition:** Nina H. Fefferman.

**Investigation:** Natalie J. Lemanski, Dina M. Fonseca.

**Methodology:** Natalie J. Lemanski, Samantha R. Schwab, Nina H. Fefferman.

**Project administration:** Nina H. Fefferman.

**Software:** Natalie J. Lemanski, Samantha R. Schwab, Nina H. Fefferman.

**Supervision:** Dina M. Fonseca, Nina H. Fefferman.

**Writing – original draft:** Natalie J. Lemanski.

**Writing – review & editing:** Natalie J. Lemanski, Samantha R. Schwab, Dina M. Fonseca, Nina H. Fefferman.

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
