## [Decision Letter · Decision Letter 0]

8 Dec 2019

Dear Dr. Lemanski:

Thank you very much for submitting your manuscript "Coordination among neighbors improves the efficacy of Zika control despite economic costs" (#PNTD-D-19-01701) for review by PLOS Neglected Tropical Diseases. Your manuscript was fully evaluated at the editorial level and by independent peer reviewers. The reviewers appreciated the attention to an important problem, but raised some substantial concerns about the manuscript as it currently stands. These issues must be addressed before we would be willing to consider a revised version of your study. We cannot, of course, promise publication at that time.

We therefore ask you to modify the manuscript according to the review recommendations before we can consider your manuscript for acceptance. Your revisions should address the specific points made by each reviewer. 

When you are ready to resubmit, please be prepared to upload the following:

(1) A letter containing a detailed list of your responses to the review comments and a description of the changes you have made in the manuscript.

(2) Two versions of the manuscript: one with either highlights or tracked changes denoting where the text has been changed (uploaded as a "Revised Article with Changes Highlighted" file); the other a clean version (uploaded as the article file).

(3) If available, a striking still image (a new image if one is available or an existing one from within your manuscript). If your manuscript is accepted for publication, this image may be featured on our website. Images should ideally be high resolution, eye-catching, single panel images; where one is available, please use 'add file' at the time of resubmission and select 'striking image' as the file type. 

Please provide a short caption, including credits, uploaded as a separate "Other" file. If your image is from someone other than yourself, please ensure that the artist has read and agreed to the terms and conditions of the Creative Commons Attribution License at http://journals.plos.org/plosntds/s/content-license (NOTE: we cannot publish copyrighted images). 

(4) If applicable, we encourage you to add a list of accession numbers/ID numbers for genes and proteins mentioned in the text (these should be listed as a paragraph at the end of the manuscript). You can supply accession numbers for any database, so long as the database is publicly accessible and stable. Examples include LocusLink and SwissProt.

(5) To enhance the reproducibility of your results, we recommend that you deposit your laboratory protocols in protocols.io, where a protocol can be assigned its own identifier (DOI) such that it can be cited independently in the future. For instructions see http://journals.plos.org/plosntds/s/submission-guidelines#loc-methods

While revising your submission, please upload your figure files to the Preflight Analysis and Conversion Engine (PACE) digital diagnostic tool, https://pacev2.apexcovantage.com/ PACE helps ensure that figures meet PLOS requirements. To use PACE, you must first register as a user. Then, login and navigate to the UPLOAD tab, where you will find detailed instructions on how to use the tool. If you encounter any issues or have any questions when using PACE, please email us at figures@plos.org.

We hope to receive your revised manuscript by Feb 06 2020 11:59PM. If you anticipate any delay in its return, we ask that you let us know the expected resubmission date by replying to this email.

To submit a revision, go to https://www.editorialmanager.com/pntd/ and log in as an Author. You will see a menu item call Submission Needing Revision. You will find your submission record there. 

Sincerely,

Sassan Asgari

Guest Editor

Serap Aksoy

Editor-in-Chief

While both reviewers have found the work and the question interesting, reviewer 2 has a major concern with the applicability of the modelling to Ae. albopictus and Ae. aegypti. Addressing this concern, in particular, and others are important in determining the suitability of the work for publication.

Reviewer's Responses to Questions

**Key Review Criteria Required for Acceptance?**

**Methods**

-Are the objectives of the study clearly articulated with a clear testable hypothesis stated?

-Is the study design appropriate to address the stated objectives?

-Is the population clearly described and appropriate for the hypothesis being tested?

-Is the sample size sufficient to ensure adequate power to address the hypothesis being tested?

-Were correct statistical analysis used to support conclusions?

-Are there concerns about ethical or regulatory requirements being met?

Reviewer #1: (No Response)

Reviewer #2: -Are the objectives of the study clearly articulated with a clear testable hypothesis stated?

Yes.

-Is the study design appropriate to address the stated objectives?

Yes

-Is the population clearly described and appropriate for the hypothesis being tested?

Yes.

-Is the sample size sufficient to ensure adequate power to address the hypothesis being tested?

Yes. 

-Were correct statistical analysis used to support conclusions?

n/a

-Are there concerns about ethical or regulatory requirements being met?

No.

-Can you describe where the parameter values were derived from or the reasoning for the chosen values?

- The parameter assumptions about proportions of larvae killed in treated/untreated areas may not be accurate. If we are saying that it is not possible to treat all larval habitats because they are cryptic and/or ephemeral, then treatment doesn’t make a difference to the larval survival or at least will only ever be partial. 

-lines 139-152: are there any examples of locations/control programs that use the epidemiologic-based surveillance that you describe, in real time? This sounds very resource-intensive and I would be surprised if anyone is able to do it.

**Results**

-Does the analysis presented match the analysis plan?

-Are the results clearly and completely presented?

-Are the figures (Tables, Images) of sufficient quality for clarity?

Reviewer #1: (No Response)

Reviewer #2: Results

-Does the analysis presented match the analysis plan?

Yes.

-Are the results clearly and completely presented?

Yes.

-Are the figures (Tables, Images) of sufficient quality for clarity?

Yes.

**Conclusions**

-Are the conclusions supported by the data presented?

-Are the limitations of analysis clearly described?

-Do the authors discuss how these data can be helpful to advance our understanding of the topic under study?

-Is public health relevance addressed?

Reviewer #1: (No Response)

Reviewer #2: Conclusions

-Are the conclusions supported by the data presented?

-Are the limitations of analysis clearly described?

No. Please elaborate on the limitations of choosing single parameter values (versus a range) and limitations related to the assumptions made in the study. 

-Do the authors discuss how these data can be helpful to advance our understanding of the topic under study?

Yes.

-Is public health relevance addressed?

Yes.

**Editorial and Data Presentation Modifications?**

Reviewer #1: (No Response)

Reviewer #2: -line 70: Guillain-Barré has an accent on the ‘e’

-line 75-76: this sentence is confusing; there is no Zika vaccine yet and there are some dengue vaccines with limited availability

-line 78: Albopictus is not capitalized, sentence thought is incomplete (immature what?)

**Summary and General Comments**

Reviewer #1: The authors present an interesting study modeling larviciding activities in different contexts on surveillance. I found the article to be well written and well described. 

1. The authors need a limitation section in the discussion, with points about the assumptions their model makes. Equal spatial distribution, finding all the breeding sites, etc.

2. The article would benefit greatly from a visual representation of the model that is being tested.

3. The abstract references surveillance, the title references control. Can the authors standardize those? I see that the paper is examining coordinated control efforts in different contexts of surveillance. But currently that is a bit confusing.

4. In the abstract line 36 can the authors spell out SIR?

5. Line 90. Perhaps revise to remove hanging preposition; “migration of biting adults from untreated patches may compensate for mortality in treated patches…”

6. Line 118. Spell out first instance of SIR.

7. Need instructions in the manuscript for accessing the data.

Reviewer #2: This paper considers the spread of Aedes aegypti and Aedes albopictus between controlled and non-controlled areas and asks how coordination between these areas might benefit disease control even under conditions where this coordination is costly in terms of time or resources. 

Lines 84-94 describe an interesting scenario and give good reason for asking the proposed research question, but it isn’t clear to me that this scenario is applicable to Aedes aegypti or Aedes albopictus. The adults of these species have a very short flight span, and it seems that only under rare occasions would these adults be traveling between controlled and non-controlled patches. If we imagine control within a city, that control is already being coordinated within the city, and these Aedes species are unlikely to travel outside of the city in their adult stage. It is hard to imagine patches of adult migration range that would be under different treatment schedules. 

Even if we think about the edges of different neighborhoods within a city that may be treated a different times for larval control, we would have to imagine a scenario under which there is no adulticide being applied or that it is being applied infrequently enough that there is a window of time during which the adults would travel between these neighborhoods and could find untreated pools in which to lay eggs. It’s not clear that those windows of time are that great (a few days at most?) within an average city in these endemic areas. 

This paper’s hypothesis would be better supported with some description of control plans in endemic areas and more background on control in general so that the reader may better evaluate the likelihood of the situations described. 

Many places use larval surveillance methods (generally cheaper and less labor-intensive than adult surveillance)…. would the results change if larval surveillance and larval control were the focus? Can you further elaborate on this limitation?

PLOS authors have the option to publish the peer review history of their article (what does this mean?). If published, this will include your full peer review and any attached files.

Reviewer #1: No

Reviewer #2: No

---

## [Decision Letter · Decision Letter 1]

30 Mar 2020

Dear Dr. Lemanski,

We are pleased to inform you that your manuscript 'Coordination among neighbors improves the efficacy of Zika control despite economic costs' has been provisionally accepted for publication in PLOS Neglected Tropical Diseases.

Best regards,

Sassan Asgari

Guest Editor

Serap Aksoy

Editorial Advisor

Reviewer's Responses to Questions

**Key Review Criteria Required for Acceptance?**

**Methods**

-Are the objectives of the study clearly articulated with a clear testable hypothesis stated?

-Is the study design appropriate to address the stated objectives?

-Is the population clearly described and appropriate for the hypothesis being tested?

-Is the sample size sufficient to ensure adequate power to address the hypothesis being tested?

-Were correct statistical analysis used to support conclusions?

-Are there concerns about ethical or regulatory requirements being met?

Reviewer #1: (No Response)

Reviewer #2: -objectives, hypothesis, and methods are adequately explained

-approach and data are appropriate

-no ethical concerns

-table of assumptions and figure of model structure are excellent additions to this section

**Results**

-Does the analysis presented match the analysis plan?

-Are the results clearly and completely presented?

-Are the figures (Tables, Images) of sufficient quality for clarity?

Reviewer #1: (No Response)

Reviewer #2: -analysis is as described

-results and figures are clear

-the sensitivity analysis was a great addition and helps inform the complex considerations for application in practice

-additional supplemental materials are appreciated

**Conclusions**

-Are the conclusions supported by the data presented?

-Are the limitations of analysis clearly described?

-Do the authors discuss how these data can be helpful to advance our understanding of the topic under study?

-Is public health relevance addressed?

Reviewer #1: (No Response)

Reviewer #2: -thank you for adding the additional discussion of limitations

**Editorial and Data Presentation Modifications?**

Reviewer #1: (No Response)

Reviewer #2: (No Response)

**Summary and General Comments**

Reviewer #1: (No Response)

Reviewer #2: This paper is well-written and informative. The additional information in the introduction and methods have improved the article and give readers important contextual information on mosquito control. The clarifications helped to understand the spatial scale of the model and how this work applies to Aedes species. Thanks you for the additions to the discussion and the extra effort for the sensitivity analysis. This work provides a helpful basis for discussion of cost-benefit efforts in vector control management.

PLOS authors have the option to publish the peer review history of their article (what does this mean?). If published, this will include your full peer review and any attached files.

Reviewer #1: No

Reviewer #2: Yes: Rachel J Sippy

---

## [Editor Report · Acceptance letter]

12 Jun 2020

Dear Dr. Lemanski,

We are delighted to inform you that your manuscript, "Coordination among neighbors improves the efficacy of Zika control despite economic costs," has been formally accepted for publication in PLOS Neglected Tropical Diseases.

Best regards,

Shaden Kamhawi

co-Editor-in-Chief

Paul Brindley

co-Editor-in-Chief
